# Levofloxacin Cocrystal/Salt with Phthalimide and Caffeic Acid as Promising Solid-State Approach to Improve Antimicrobial Efficiency

**DOI:** 10.3390/antibiotics11060797

**Published:** 2022-06-13

**Authors:** Noor Ul Islam, Muhammad Naveed Umar, Ezzat Khan, Fakhria A. Al-Joufi, Shaymaa Najm Abed, Muhammad Said, Habib Ullah, Muhammad Iftikhar, Muhammad Zahoor, Farhat Ali Khan

**Affiliations:** 1Department of Chemistry, University of Malakand, Chakdara, Dir Lower 18800, Khyber Pakhtunkhwa, Pakistan; nooruomchem@gmail.com (N.U.I.); m.naveedumar@uom.edu.pk (M.N.U.); ekhan@uom.edu.pk (E.K.); msaidqau@yahoo.com (M.S.); 2Department of Chemistry, College of Science, University of Bahrain, Sakhir 32038, Bahrain; 3Department of Pharmacology, College of Pharmacy, Jouf University, Aljouf 72341, Saudi Arabia; faaljoufi@ju.edu.sa; 4Nursing Department, College of Applied Medical Sciences, Jouf University, Sakaka 72311, Saudi Arabia; snabed@ju.edu.sa; 5Department of Nuclear and Quantum Engineering, Korea Advanced Institute of Science and Technology (KAIST), Daejeon 34141, Korea; habibuom24@gmail.com; 6Department of Biochemistry, University of Malakand, Chakdara, Dir Lower 18800, Khyber Pakhtunkhwa, Pakistan; miftikhar355@gmail.com; 7Department of Pharmacy, Shaheed Benazir Bhuto University, Sheringal, Dir Upper 18000, Khyber Pakhtunkhwa, Pakistan; farhatkhan2k9@yahoo.com

**Keywords:** levofloxacin/phthalimide, levofloxacin/caffeic acid, heterosynthons, antibacterial activity, MDR strains

## Abstract

To overcome the issue of multidrug resistant (MDR) microbes, the exploration of ways to improve the antimicrobial efficiency of existing antibiotics is one of the promising approaches. In search of synthons with higher efficiency, in current investigations, cocrystal and amorphous salt of levofloxacin hemihydrate (LEV) were developed with phthalimide (PTH) and caffeic acid (CFA). New materials were characterized with the help of FT-IR, Raman spectroscopy, powder X-ray diffraction (PXRD), differential scanning calorimetry (DSC) and thermogravimetric analysis (TGA). Shifting, attenuation, appearance/disappearance and broadening of bands were observed in the FT-IR and Raman spectra of the materials as evidence of the required product. The PXRD diffraction pattern observed for LEV-PTH indicated cocrystal while halo diffractogram of LEV-CFA revealed amorphous nature. DSC/TG analysis confirmed the hydrated nature of the cocrystal/salt. The dissolution rate and antimicrobial activity against selected strains, *K.*
*pneumonia*, *E. coli* and *S. typhi* of parent drug and the new material were compared. The zone of inhibition (ZI) observed for 5 µg LEV-PTH was 30.4 + 0.36 (*K. pneumonia*), 26.33 + 0.35 (*E. coli*) and 30.03 + 0.25 mm (*S. typhi*) while LEV-CFA salt (5 µg) against the same strains inhibited 33.96 ± 0.25, 31.66 ± 0.35 and 27.93 ± 0.40 mm, respectively. These novel formulations enhance the dissolution rate as well as antibacterial efficiency and are expected to be potent against MDR bacterial strains.

## 1. Introduction

Solid state properties of APIs can be improved via salt [1,2], cocrystal [3,4] and coamorphous [5,6,7] conversions. Such multicomponent solid formulations containing an API and a coformer in a well-defined stoichiometric ratio and interact through non-covalent interactions. Cocrystallization [8,9], designing of salt [10,11,12] and coamorphization [5,6,7] approaches are preferably used to enhance the solubility, dissolution rate and permeability. Recently such approaches have been used for taste masking [13,14], photostabilization [15,16,17], color tuning [18] and improving antimicrobial properties [8,12,19,20]. Cocrystals/salts of ciprofloxacin with thymol, carvacrol [20], hippurate [12] and polymers [19] have been successfully prepared. The cocrystals/salts exhibit better antibacterial efficiency than the respective pure drug [20]. Lima et al. tested synergistic antibacterial effect of caffeic acid (CFA) with antibiotics norfloxacin, imipenem and gentamicin tested against three bacterial strains *Staphylococcus aureus* (*S. aureus*), *Escherichia coli* (*E. coli*) and *Pseudomonas aeruginosa* (*P. aeruginosa)*. They observed significant improvement in antibacterial spectrum of the tested antibiotics [21]. Phthalimide (PTH) is usually employed in the design of potential anti-tumor [22], immunemodulatory [23], anti-angiogenic [24], anti-microbial [25] and anti-inflammatory [26] drugs. The Lamotrigine-PTH and venlafaxine-CFA cocrystal/salt have also been reported in literature which showed better performance than respective pure drug [27].

Levofloxacin hemihydrate (LEV hereafter) is a broad-spectrum antibiotic which is a third-generation fluroquinolone. It is the *levo* isomer of oflaxacin [28,29,30] soluble up to 25 mg/mL in water and having bitter taste [31]. According to the biopharmaceutical classification system, it has been classified as a class-I drug which has no severe solubility or bioavailability problems [32]. It is absorbed well from the gastrointestinal tract, and maximum peak plasma concentration is reached within 1 to 2 h when administered orally under fasting conditions [13]. It is available in different formulations that can be administered orally, in the form of eye drops and intravenously. Though LEV has been used globally, antimicrobial resistance against LEV has also surfaced in several studies, which reduced its efficacy [33,34]. Two hydrates (monohydrate and hemihydrate) forms and four anhydrous polymorphs have been reported in currently available literature. The anhydrous polymorphs quickly absorb water and get hydrated which changes the physicochemical properties of the formulation. Data on their changed physicochemical properties in the literature are scarce. The crystal structure of hydrated LEV has been well documented [32,35,36] while two anhydrous polymorphs structure have been recently determined using high-temperature PXRD [37]. Cocrystal, which was superior to hydrated forms of LEV in both pharmacological and physicochemical properties, can be a promising API candidate. Cocrystal of LEV and metacetamol exhibited better physicochemical properties, including as hygroscopicity, physical stability, and photostability, while keeping good dissolving characteristics and chemical stability under a variety of temperature and humidity circumstances [15]. Bandari et al. [13] used a solvent evaporation approach to create LEV cocrystals, resulting in taste masking and increased LEV dissolution rate. Singh et al. [32] reported that they obtained several salts by solvent-drop grinding while screening for LVFX coformers.

The emergence and spread of antimicrobial resistance have been of serious concern to human health and the management of bacterial infectious diseases [38]. The good health benefits that have been accomplished with antibiotics lurk due to speedily evolving resistant bacteria. The lack of new antimicrobial agents development [39] and bacteria that were susceptible to antibiotics are receiving tolerant to these drugs [40] have been serious concerns globally. In recent years, AMR has intensified which is a threat for public health globally [41]. Intensive overuse of antibiotics to control infections in human, animals and agriculture is the main reason for drug resistance development in microbes [42]. Actually, the reason AMR is a significant concern is the high mortality level attributed to infections caused by multi drug resistant germs [43]. A number of common pathogenic strains already bear antibiotic-resistant genes, and presumably, more antibiotic-resistant pathogens will emerge in the future, if no different and more cautious use of antimicrobials takes place [44]. Two possible approaches can be used to deal with the antimicrobial resistance (AMR): the first is the development of novel active materials, but it faces increasing bench-to-market costs and times [45]; the second is the investigation of ways to enhance the efficacy of current antimicrobials, and it is more favorable [46]. In this investigation, cocrystallization/salt strategies were applied to cope with the AMR problem, as the AMR is currently one of the burning issues in most healthcare systems worldwide [47]. Based on the improving antimicrobial potencies of antibiotics through synthesis of cocrystals/salts and pharmaceutical applications of coformers (PTH and CFA), the LEV cocrystal/salt with the coformers was prepared to improve antibacterial and dissolution rates. The work developed in the present study shows unprecedented importance, especially in the case of the evaluation of antimicrobial modulation through the synthesis of cocrystal/salt. Therefore, the objective of this work was to investigate biological properties of the resulting formulations, including dissolution rate and antimicrobial activities.

## 2. Results and Discussion

### 2.1. Theoretical Considerations

The chemical structures of LEV and coformers (PTH and CFA) are displayed in Figure 1. The LEV molecule contains one hydrogen bond donor (O–H group of carboxylic acid) and seven potentially hydrogen bond acceptors (three N-atoms and four O-atoms) [32]. LEV and coformers (CFA and PTH) structures presented in Figure 1 evidence the presence of active groups (OH, COOH, piperazine ring and NH) which may interact and form a cocrystal/salt. The hydrogen bond donor/acceptor (OH) and (C=O) groups of LEV and imide group of PTH would form heterosynthon with the carboxylic group as previously reported for levofloxacin-saccharine [13] and indomethacin-saccharine [48,49] cocrystal systems, while CFA would possibly form homosynthon with the carboxylic group of the drug [13]. LEV may also form homodimer synthon which further interacts with amide group of PTH like indomethacin-saccharin [48] and carbamazepine-saccharin cocrystals [50]. In contrast, the OH group of LEV is involved in intermolecular hydrogen bonding with adjacent carbonyl oxygen-atom which makes them unavailable for establishing supramolecular structure [15,32]. However, the carboxy oxygen of the carboxylic group acts as a hydrogen bond acceptor and establishes hydrogen bond with the NH (amide) group of metacetamol [15]. The absence of strong hydrogen bond donor groups and a high acceptor/donor ratio also makes it susceptible for hydration under normal conditions. The N-methylpiperazine group accepts protons from several coformers and forms molecular salts [32]. Shinozaki et al. report a cocrystal of LEV whereas nitrogen of the methylpiperazine ring was found to be involved in hydrogen bond formation [15]. Therefore, the active interaction sites of PTH and CFA have also the possibility to interact with N of methylpiprizine ring to form supramolecular synthons. Furthermore, in lamotrigine-PTH cocrystals the amide part of PTH interacts with the aminopyridine of lamotrigine [27] while transfers of protons from the carboxylic group of CFA to the tertiary amino group of venlafaxine have taken place in venlafaxine-CFA salt [51].

Generally, it is accepted that the reaction of base (in our case LEV) with acid (in our case PTH and CFA) is projected to form a salt if ΔpKa = pKa(base) − pKa(acid) > 2 or 3 and a smaller ΔpKa (less than 0) will form cocrystal exclusively [10,52,53,54]. We can also describe equilibrium phenomena of solution from pKa values which is to be a useful parameter to predict preliminary the state of ionization in solid state (salt/cocrystal) [53,54,55,56,57].
LEVH^+^ ⇌ LEV + H^+^     pKa = 7.94 
PTH ⇌ PTH^−^ + H^+^ pKa = 8.3
CFA ⇌ CFA^−^ + H^+^     pKa = 4.37
For reaction, PTH + LEV ⇌ PTH^−^ + LEVH^+^
K_eq_ = [PTH^−^][LEVH^+^]/[PTH][LEV] = 10^7.94^/10^8.3^ ≈ 0.436
For reaction CFA + LEV ⇌ CFA^−^ + LEVH^+^
K_eq_ = [CFA^−^][LEVH^+^]/[CFA][LEV] = 10^7.94^/10^4.37^ ≈ 3.717 × 10^3^

The equilibrium equations of LEV and PTH indicate that the concentration of ionized species is less than that of unionized species in the solution containing the same molar ratio while the equimolar solution containing LEV and CFA revealed that ionized species are 3.717 × 10^3^ greater in numbers than nonionized species. The cocrystal formation of LEV-PTH and salt formation of LEV-CFA is consistent with the ΔpKa rules (also called rule of 3).

### 2.2. Construction of Binary Phase Diagram and Determination of Melting Point

A binary phase was constructed to predicate an appropriate ratio of drug-coformer for preparation of cocrystal/salt [58,59]. Melting point data of physical mixtures with different mole ratios were plotted against melting temperature, as shown in Appendix A. The “W” shape binary phase diagrams of LEV with PTH and CFA indicate that 1:1 molar ratio leads the formation of cocrystal/salt [58,59,60,61]. Additionally, the melting point of respective pure materials, LEV-PTH cocrystal and LEV-CFA salt are presented in Table 1. The melting points of cocrystal and salt are different than drug and respective coformers which provided preliminary evidence regarding the successful formation of the cocrystal/salt in 1:1 molar ratio [12].

### 2.3. FT-IR Analysis

The FT-IR spectra of LEV, PTH, CFA and the resulted cocrystal/salt are presented in Figure 2. The bands of LEV [13,15], PTH [62] and CFA [63,64] were assigned based on previously published data. LEV showed characteristics bands in the range of 3450 to 1100 cm^−1^. The broad band at 3252 cm^−1^ corresponds to the stretching vibration of O-H, while 2972, 2934 cm^−1^ corresponds to the stretching vibrations of methyl groups. The C=O and C-N stretching vibration bands of LEV appeared at 1722 and 1087 cm^−1^, respectively. The N-H stretching band of PTH appeared at 3190 cm^−1^, while NH in-plane and out of plane bands were observed at 1380 and 1055 cm^−1^. The C=O bands appeared at 1775 and 1722 cm^−1^, two bands 1354 and 1307 cm^−1^ appeared due to C-N vibration. The range 1603–1407 cm^−1^ is attributed to C=C vibration. The very strong bands at 1645 and 1642 cm^−1^ in the CFA spectrum were assigned to the C=O stretching modes of the carboxylic group. Bands appearing at 3399 and 3222 cm^−1^ correspond to the COOH and OH groups, respectively. In addition, the bands of the very strong and medium intensities in the IR spectrum at 1620, 1524, and 1450 cm^−1^ were assigned to the CC stretching modes of both benzene moiety and acyclic chain.

An FT-IR spectroscopic study was performed to identify the non-covalent interactions within the cocrystal/salt. A shift in the carbonyl group of the acid or amide derivatives is common when the group is involved in intermolecular interactions [65,66]. In LEV-PTH cocrystal spectrum the N-H and two C=O bands shifted to lower wavelength 3180, 1770 and 1711 cm^−1^, respectively. In LEV-CFA salt the CFA COOH group bands disappeared due to deprotonation whereas the OH peak disappeared due to its involvement in hydrogen bonding, these bands disappeared in the reported FTIR spectrum of venlafaxine-CFA salt [51]. Additionally, the COO^−^ bands appeared at 1524, 1375 and 702 cm^−1^ which suggests a deprotonated form of CFA [64] in LEV-CFA. Furthermore, the CFA carbonyl band in the LEV-CFA salt spectrum appeared at a different position with low intensity while a broad band formed in the tentative range 2700–2400 cm^−1^ due to protonated piperazine nitrogen (NH^+^). Different FT-IR patterns (bands attenuation, shifting and disappearance) of LEV-PTH and LEV-CFA than respective parental material particularly NH, OH and C=O bands lead to the conclusion that cocrystal/salt is formed due to intermolecular interactions [67]. The pka_1_ (carboxylic group) and pKa_2_ (piperazine) have been reported 5.59 and 7.94, respectively [55]. The pKa difference (−0.9) between the drug (piperazine) and PTH (amide; pKa 8.3) [56] falls within the limits of cocrystal formation while the pKa difference (3.34) between the piperazine and carboxylic acid of CFA (carboxylic group; pKa 4.37) [57] supports salt formation. Therefore, based on FT-IR analysis and ΔpKa rule, it is suggested that LEV-PTH and LEV-CFA afford cocrystal and salt, respectively.

### 2.4. Raman Analysis

The Raman spectra of LEV, PTH, CFA, and formulations (LEV-PTH and LEV-CFA) are given in Figure 3. The peaks of LEV-PTH appeared with low intensity and slight shifting in comparison with parental materials. In the spectrum of LEV-CFA salt, the prominent peaks of both raw materials were observed with low intensity. Additionally, some peaks disappeared, which may be due to non-covalent interactions. Many researchers have previously reported attenuation, broadening, shifting and appearing/disappearing of Raman bands in cocrystals/salts spectra [68,69,70,71,72]. Based on literature data for cocrystals/salts, it is suggested that LEV-PTH and LEV-CFA interact with each other, which is the requirement for successful preparation of cocrystal/salt.

### 2.5. PXRD Analysis

For detailed structural information, crystals with appropriate dimensions and sizes have to be prepared, which is often a hectic task and fails in many instances. Another limitation of single crystal analysis is that the selected crystal may represent a side product, not the desired polycrystalline product (bulk). Moreover, single crystal analysis is a time consuming and not readily available technique. Conversely, PXRD is a readily available technique generally used for confirmation and determination of bulk purity and crystallinity of the bulk material [73]. Reference X-ray diffraction patterns should be those calculated from single crystal X-ray diffraction measurements. However, it is difficult to get suitable single crystals for single crystal X-ray diffraction to get the data. Therefore, we selected the X-ray diffraction patterns of the powder crystalline material for cocrystal as the reference. Crystals were crushed before measurements to eliminate the effects of preferential orientation and enhance the clarity of the small diffraction peaks [74,75]. The diffractograms of LEV, PTH, CFA and formulations (LEV-PTH and LEV-CFA) are given in Figure 4. The LEV high intensity peaks showed at 2-theta (deg) values 6.29°, 9.93°, 12.91°, 15.50°, 19.03° and 26.3° while PTH showed sharp diffraction peaks at 8.1°, 13.86°, 15.50°, 23.77°, 26.60°, 27.07° and 28.75°. The diffraction peaks of CFA observed at 2-theta (deg) and PDA peaks were noted. The diffraction peaks of starting material LEV [13,15], PTH [27] and CFA [51,76] show good consistency with previously compiled values. New diffraction peaks appeared at 11.50°, 20°, 29.699° and 44.2° while LEV peaks (6.29° and 26.36°) and some of the PTH peaks disappeared in the diffractogram of LEV-PTH cocrystal. Moreover, the peaks intensity pattern of the cocrystal is different from that of the respective parental materials. On the other hand, the LEV-CFA diffractogram showed no distinctive peaks. The diffractogram pattern of the LEV-PTH cocrystal in terms of position, appearance/disappearance and intensity of peaks is different than parental materials, suggesting the formation of a new crystalline phase [13,15], while the halo diffractogram of LEV-CFA revealed an amorphous nature [77].

### 2.6. DSC Analysis

The cocrystalline nature of the solid can be characterized by their melting peak while characteristic glass transition temperature revealed an amorphous system [77,78,79]. The DSC thermograms of cocrystal/amorphous and respective pure material are shown in Figure 5. Crystalline LEV, PTH and CFA exhibited melting peaks at 230, 234 and 220 °C, respectively, which shows good agreement with previously reported values [13,80,81]. The thermogram of the cocrystal showed a broad peak with maximum intensity at 97.21 °C. This peak may be attributed to bound and unbound solvate molecules. The desolvation is followed by a small endothermic peak (170 °C) and two strong endothermic peaks (174 and 226 °C). The obtained thermogram of LEV-PTH is characteristic of a cocrystal, as it is different from the V-shaped one, specific for eutectic mixture and respective starting materials [67]. Moreover, a lower endothermic peak can also be observed in an eutectic system, but the vibrational shifts in FT-IR and Raman and unique PXRD diffractogram confirm cocrystal formation. The DSC thermogram of LEV-CFA salt showed broad peaks, which are attributed to water/solvent molecules, while it underwent a glass transition event at 54 °C followed by a recrystallization peak and melting peak at 104 and 148 °C, respectively. After the melting peak, the observed peaks correspond to decomposition [78]. Considerably different thermal behaviors of LEV-PTH and LEV-CFA than those observed for the respective pure materials suggested a new solid form formation where parent molecules interact with each other.

### 2.7. TG Analysis

Thermal stability and hydration of the formulations were evaluated by TGA and thermograms are presented in Appendix A. The thermograms revealed a transformation into anhydrous form with increasing temperature. LEV-PTH cocrystal showed a gradual mass loss (2.60%) up to 90 °C, which corresponds to solvate molecules and solvent residues, while no considerable amount of mass loss was observed from 90 to 175 °C. A small amount of mass loss between 175–225 °C and onward steep mass loss were observed, which is attributed to decomposition of the cocrystal. In the case of LEV-CFA salt, a 6.6% mass loss was observed up to onset temperature (147 °C) which indicates the presence of solvate and solvent residues. This is to be expected for amorphous formulations due to their hygroscopic nature. Beyond the onset temperature, a substantial mass loss of the amorphous salt occurred which corresponds to decomposition. The decomposition of amorphous salt began at a lower temperature than the corresponding pure constituents. This may be due to the higher molecular mobility of the amorphous LEV-CFA, which can increase their reactivity and thus make them more prone to chemical degradation [82].

### 2.8. ^1^H-NMR Studies

NMR spectroscopy is of immense importance in elucidating the structure of compounds. In the field of cocrystals, it helps to determine the presence of molar ratio of various components in heterosynthons. The ^1^H-NMR data of LEV-CFA indicate a very broad peak with a shoulder in downfield at 15.18 ppm which can be assigned to OH groups of the molecules. There are some unambiguous peaks in both fractions which can be assigned with greater certainty to the respective protons. The vinyl hydrogens in CFA are trans to each other and give duplet peaks at 7.42–7.39 and 6.15–6.19 ppm with a 3J (^1^H,^1^H) coupling equal to 15.9 Hz, while one of the ring protons of LEV gives a singlet in the low field at 8.95 ppm. Integration of these protons is exactly in a 1:1 ratio, indicating that components in bulk amount of the material are present in equimolar ratios. Other signals of the compound are within the expected range, given in Appendix A. The ^1^H-NMR of LEV-PTH is not as straightforward as that of LEV-CFA. There are four aromatic and one NH protons in PTH. The latter is broader, and it does not give exact value integration, and aromatic protons appear very close to each other in the aromatic region as multiplets. An overview of the spectrum reveals that both synthons are present in a 1:1 ratio. NMR data provides enough insights regarding the molar ratio of individual components present in the bulk material.

### 2.9. In Vitro Powder Dissolution Study

The dissolution profiles of LEV-PTH and LEV-CFA formulations were evaluated in salivary pH 6.8 (phosphate buffer) and simulated gastric fluid pH 1.2 (free enzyme), and the results are displayed in Figure 6. The result obtained revealed that the dissolution profile of LEV-PTH cocrystal in salivary pH is lower than pure LEV, while LEV-CFA amorphous salt exhibited better performance than the respective drug. On the other hand, the dissolution profile of the cocrystal and amorphous salt in simulated gastric fluid improved as compared with the pure drug.

### 2.10. In Vitro Antimicrobial Study

The improvement of antibacterial activity against bacterial strains through cocrystallization [8,20,83,84,85] and amorphous salt [19] has been reported in literature. Therefore, the comparative zone of inhibition and minimum inhibitory concentration (MIC) studies of LEV and respective formulations were performed against bacterial strains *Escherichia coli* (*E. coli*), MTCC 1687; *Salmonella typhi* (*S. typhi*), MTCC 734 and *Klebsiella pneumonia* (*K. pneumonia*), MTCC 1030; the results have been presented in Table 2 and Table 3. LEV-PTH and LEV-CFA showed better performance than the parental drug despite a lower amount of LEV being present in drug formulations. The improvement of antibacterial activity can be explained as CFA exhibited an antimicrobial as well as synergistic effect with antibiotics [21] while PTH have also been used for designing antimicrobial agents [25]. Moreover, enhancement of antimicrobial activity of fluroquinolone drug (ciprofloxacin) cocrystals with natural preservatives has also been reported in the literature [20]. The activity of the prepared formulations in terms of bacterial inhibition was improved; the technique is promising and can be modified for MDRS (multidrug resistant strains) in future studies.

## 3. Experimental Section

### 3.1. Materials

LEV powder was obtained from local pharmaceutical industry and used without further purification. Distilled water was purchased from local market while rest of the reagents and solvents used in this work were of HPLC grade and purchased from Sigma-Aldrich.

### 3.2. Cocrystal/Salt Synthesis

LEV cocrystal/salt was synthesized using a solvent evaporation method. LEV and coformers (PTH and CFA) solutions were prepared in equimolar (1:1) ratio using a mixture of solvents (water: methanol, *v/v* 50:50%). The solutions were sonicated for 15 min at 60 °C, and volatiles were allowed to evaporate. Crystalline material of LEV-PTH and glass like material of LEV-CFA were obtained and stored for further studies.

### 3.3. Construction of Binary Phase Diagram and Determination of Melting Point

Physical mixtures (LEV-PTH and LEV-CFA) of different mole ratios were grinded for 15 min using pestle and mortar. The melting points of physical mixtures were measured using a thermal apparatus (Bibby Scientific Limited Stone, Staffordshire, ST15 OSA, UK). Additionally, the melting points of LEV-PTH, LEV-CFA and respective pure materials were also measured.

### 3.4. Characterization of Cocrystal/Coamorphous

Cocrystals/salt was characterized by vibrational spectroscopy (FT-IR, Raman), PXRD, thermal analysis (DSC and TGA) and ^1^H-NMR. The FT-IR analyses, 4000–500 cm^−1^ at 2 cm^−1^ spectral resolution with the accumulation of 256 spectral scans were performed using an FT-IR (perkinelmer spectrum−10.5.1) spectrophotometer. The sample placed on the sample holder directly and the IR spectrums of the sample are obtained on the computer screen. Raman spectra were obtained (1800–200 cm^−1^) using Lab RAM HR, Horiba Jobin Yvon, France, operated at a resolution of 5 cm^−1^ and using a laser wavelength of 785 nm. The data were acquired using front-face scattering from a thick powder bed contained in an aluminum sample holder. The PXRD analyses were performed using an EQUINOX 3000 X-Ray diffractometer with CPS-120 detector, thermo scientific company, Cu-Kα radiation, wavelength 1.54056 Å. The samples were scanned at 2θ from 5–50° with time 0.05 s per scan. Thermal responses of cocrystal/salts, drug and corresponding coformers were measured by DSC-60 (Shimadzu, Japan). The STARe software was used for data processing and analysis. Thermal gravimetric studies were performed with the help of Diamond Series TG/DTA Perkin Elmer, USA, using Al_2_O_3_ as reference. 1H-NMR spectra of both the products were measured in deuterated solvents and room temperature using Bruker AVANCE spectrophotometer (400 MHz).

### 3.5. In Vitro Powder Dissolution Study

Dissolution study of cocrystal/salt and parental drug were performed using USP apparatus type-2 (paddle method). Samples were converted into fine powders and sieved through 100-mesh sieves to reduce effect of size on dissolution rate. Exactly 30 mg of samples were added to the vessels of apparatus having 900 mL phosphate buffer (pH 6.8). The rotation speed was adjusted to 75 rpm at 37 °C. Aliquots of 5 mL were withdrawn at predetermined time intervals up to 60 min. After withdrawing aliquots, the same volume of dissolution medium was added to the vessel. The samples were filtered, and drug releases were analyzed by UV-spectroscopy. The same dissolution procedure was followed for a 0.1 mM HCl dissolution medium. Since coformers interfere with the *λ_max_* of LEV, multiple component analysis was used for concentration measurement.

### 3.6. In Vitro Antibacterial Study

Petri dishes and agar solution were sterilized in autoclave. A known volume (20 mL) of agar solution under sterile condition was added to each petri dish and incubated for 24 h. The plates with no contamination were used in further studies. The bacteria strains *E. coli*, MTCC 1687; *S. typhi*, MTCC 734 and *K. pneumonia*, MTCC 1030 were inoculated to plates with the help of sterile cotton swabs. The drug and equivalent cocrystal/salt (1000 µg/mL) solutions were prepared separately and further diluted to different concentrations (500, 250, 125 and 62.5 µg/mL). From each solution, 5 µL volumes were added to 6 mm disk filter paper. The plates were incubated at 37 °C for 24 h, and diameters of zone of inhibition were manually noted.

The minimal inhibitory concentration (MIC) of LEV-PTH, LEV-CFA and pure LEV were also evaluated against selected bacterial strains using the broth dilution method. Twofold serial dilutions of pure active pharmaceutical ingredient (API) and prepared formulations were made in sterile nutrient to give concentrations ranging from 2–256 μg/mL. For each MIC experiment, ten sterile tubes were labeled each 1 through 8, along with a negative control and a positive control. In separate test tubes, about 1 mL of selected strain suspensions (10^6^ CFU/mL) and 1 mL of different concentration solutions were added. The inoculated tubes were incubated at 37 °C for 24 h after which they were inspected for turbidity. A positive control (growth) was formed by culture broth with microorganisms while the negative control (sterility) consisted of broth with no microorganisms.

## 4. Conclusions and Future Work

In the current study, cocrystal/salt LEV-PTH/LEV-CFA was prepared. The PXRD diffractograms confirmed their crystalline/amorphous natures which were further confirmed by DSC. The FT-IR and Raman studies exhibited the non-covalent interactions established between LEV and the respective coformers. The dissolution rate of parental drug was improved through cocrystal/salt formation. The antimicrobial efficiency of formulations was high in comparison to parental drug against the selected bacterial strains. However, in vivo confirmations are required to exploit their applications in the field of pharmacy.

## Figures and Tables

**Figure 1 antibiotics-11-00797-f001:**
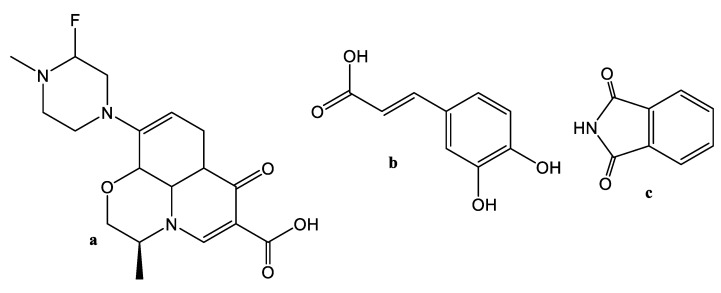
Chemical structures of LEV (**a**), CFA (**b**), and PTH (**c**).

**Figure 2 antibiotics-11-00797-f002:**
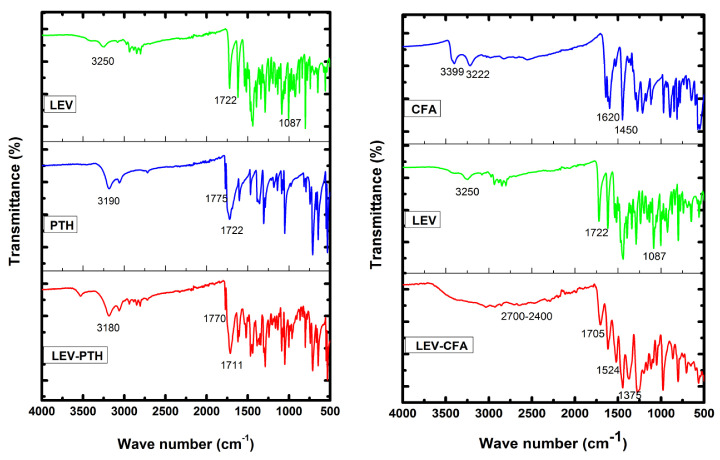
FTIR spectra of cocrystals (LEV-PTH) and amorphous salt (LEV-CFA) and respective pure components wave number (4000–500 cm^−1^).

**Figure 3 antibiotics-11-00797-f003:**
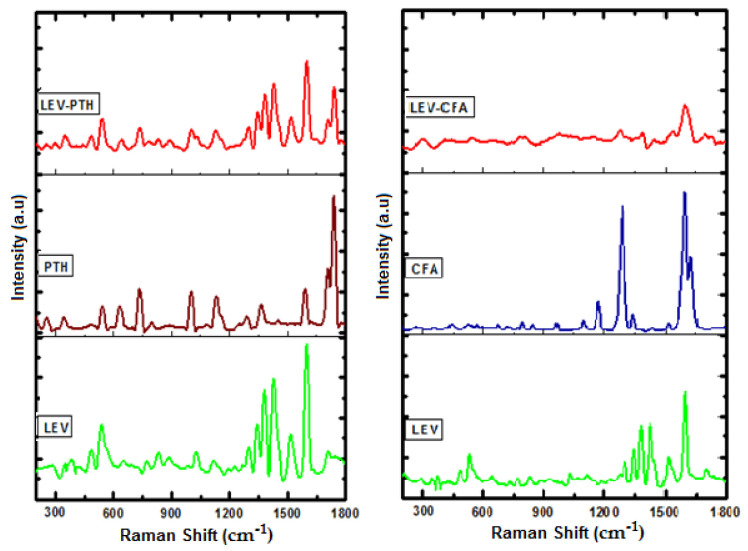
Raman spectra of cocrystal (LEV-PTH), amorphous salt (LEV-CFA) and respective pure components wave number range (200–1800 cm^−1^).

**Figure 4 antibiotics-11-00797-f004:**
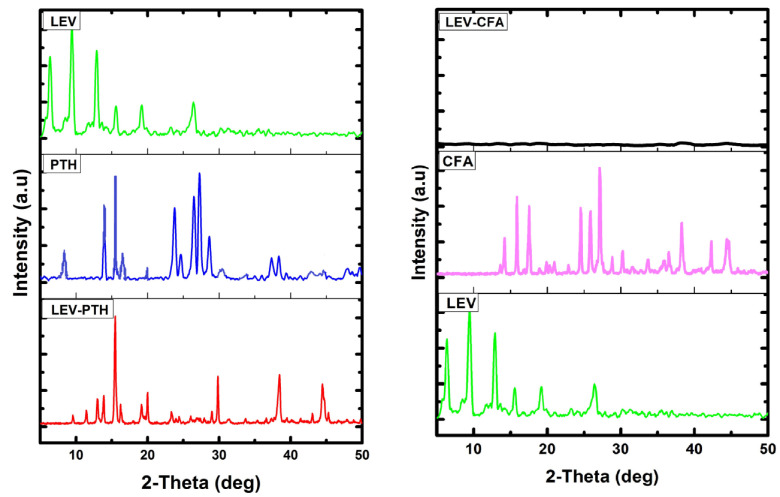
PXRD diffractograms of cocrystal (LEV-PTH), amorphous salt (LEV-CFA) and respective pure components.

**Figure 5 antibiotics-11-00797-f005:**
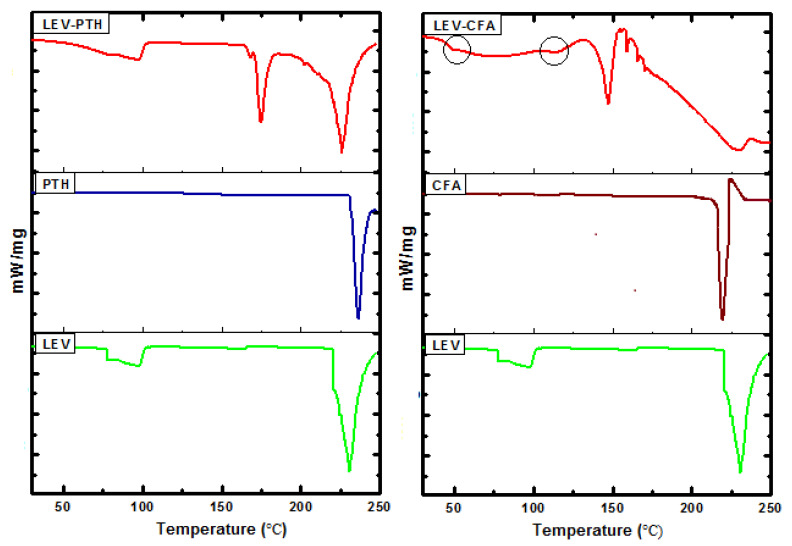
DSC thermograms of cocrystal (LEV-PTH), amorphous salt (LEV-CFA) and their respective pure components.

**Figure 6 antibiotics-11-00797-f006:**
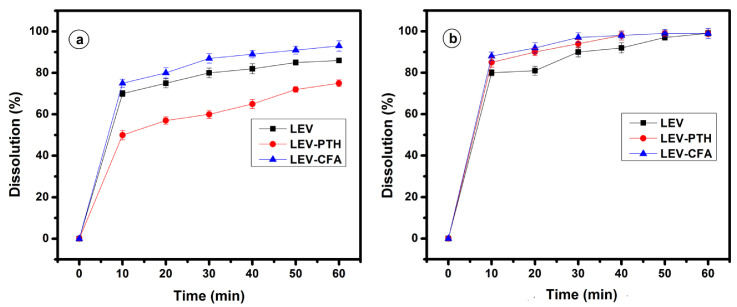
Powder dissolution profile of LEV, cocrystal (LEV-PTH) and amorphous salt (LEV-CFA) in dissolution mediums phosphate buffer pH 6.8 (**a**) and 0.1 mM HCl (**b**).

**Table 1 antibiotics-11-00797-t001:** Melting point of raw materials and cocrystal.

Sample	Melting Point (°C)
Levofloxacin	224–226
phthalimide	234–236
Caffiec acid	220–224
Cocrystal (LEV-PTH)	172–176
Salt (LEV-CFA)	150–152

**Table 2 antibiotics-11-00797-t002:** Zone of inhibition values of LEV, cocrystal and amorphous salt against different bacterial strains.

Bacterial Strain	Sample Amount (µg)	LEV (100%) Zone of Inhibition (mm)	LEV-PTH Cocrystal (71%) Zone of Inhibition (mm)	LEV-CFA Amorphous Salt (66.7%) Zone of Inhibition (mm)
*K. pneumonia*	5	28.16 ± 0.76	30.4 ± 0.36	33.96 ± 0.25
	2.5	23.33 ± 0.57	24.16 ± 0.47	28.30 ± 0.26
	1.25	17.63 ± 0.77	19.93 ± 0.60	21.76 ± 0.25
	0.62	12.26 ± 0.30	14.36 ± 0.35	18.00 ± 0.30
*E. coli*	5	25.033 ± 0.25	26.33 ± 0.35	31.66 ± 0.35
	2.5	18.13 ± 0.32	21.3 ± 0.30	26.23 ± 0.25
	1.25	14.36 ± 0.40	15.43 ± 45	21.30 ± 0.30
	0.62	12.3 ± 0.26	14.03 ± 15	18.23 ± 0.20
*S. typhi*	5	27.03 ± 0.65	30.03 ± 0.25	32.80 ± 0.20
	2.5	22.16 ± 0.37	22.93 ± 0.50	27.93 ± 0.40
	1.25	17.96 ± 0.15	19.30 ± 30	22.73 ± 0.25
	0.62	11.06 ± 0.20	14.26 ± 25	17.03 ± 0.25

Percent concentration of active antibiotic in the formulations present in the bracket in first row of column two, three and four of the table.

**Table 3 antibiotics-11-00797-t003:** MIC values of pure LEV, cocrystal and amorphous salt against different bacterial strains.

Sample	MIC (μg/mL)		
	*E. coli*	*S. typhi*	*K. pneumonia*
Pure LEV (100%)	32	128	64
LEV-PTH cocrystal (71%)	16	64	32
LEV-CFA salt (66.7%)	16	16	16

In the bracket in first column of the table is percent concentration of active antibiotic in the formulations.

## Data Availability

Data is contained within the article.

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
