# Peer review of "Levofloxacin Cocrystal/Salt with Phthalimide and Caffeic Acid as Promising Solid-State Approach to Improve Antimicrobial Efficiency"

_antibiotics, 2022, doi:10.3390/antibiotics11060797_

Round 1

Reviewer 1 Report

  1. Could you specify the measurement conditions of the Raman and FTIR spectra?
  2. Which is the rationale for non-covalent interactions to decrease intensities in Raman spectra?
  3. The LEV-PTH interaction does not present non-covalent interactions? Why is it that the Raman spectra present greater definition compared to the LEV-CFA interaction?
  4. What culture medium was used for the antimicrobial experiments?
  5. Whay you only choose bacteria from Enterobacteriaceae family
  6. What could be the effect of using a gram positive bacteria in this study
  7. I would like the figures of the antibiogram test  

Author Response

Reviewer 1

  1. Could you specify the measurement conditions of the Raman and FTIR spectra?
  • The FT-IR analyses, 4000-500 cm-1 at 2 cm-1 spectral resolution were performed using FT-IR (perkinelmer spectrum-10.5.1) spectrophotometer. The sample was placed on the sample holder directly and the IR spectrums of the sample were obtained on the computer screen.
  • Raman spectra were obtained (1800-200 cm-1) using Lab RAM HR, Horiba Jobin Yvon, France operated at a resolution of 5 cm-1 and using a laser wavelength of 785 nm.. The data were acquired using front-face scattering from a thick powder bed contained in an aluminum sample holder.
  • The same changes have also been made in revised manuscript.

  1. Which is the rationale for non-covalent interactions to decrease intensities in Raman spectra?

  • Raman spectroscopy is particularly useful for the characterization of the products and is used to determine the nature of the interactions in the cocrystals. It was observed that little change in the vibrational modes associated with the phenyl groups of the respective reactants took place upon cocrystal formation but changes in intensities of the vibrational modes associated with the amide and the carboxylic acid groups were observed upon cocrystal formation.
  • During the formation of a CIT–Pa cocrystal the (C=O), (COOH) and (NH) bands of citric acid and paracetamol are shifted to higher or lower wavenumbers by 8 to 23 cm-1 accompanied by corresponding decreases in the band intensities; which suggest that the molecular complex of citric acid and paracetamol is a cocrystal and not simply a physical mixture of these components.

Reference: Elbagerma, M.A.; Edwards, H.G.M.; Munshi, T.; Scowen, I.J. Identification of a New Cocrystal of Citric Acid and Paracetamol of Pharmaceutical Relevance. CrystEngComm 2010, 13, 1877–1884.

  1. The LEV-PTH interaction does not present non-covalent interactions?
  • Worthy reviewer, in literature during non-covalent interaction (cocrystal) very small changes up to 5 cm-1 in FTIR analysis and up to 3 cm-1 in Raman analysis are observed. The same changes have been observed over here in our study observed for LEV-PTH cocrystal. Additionally, the PXRD diffractgram and DSC thermogram also exhibited unique pattern than starting materials and it can only be possible if both LEV and PTH interacts with each other. So, overall inspection of different characterization techniques suggested non-covalent interaction.

  1.  Why is it that the Raman spectra present greater definition compared to the LEV-CFA interaction?
  • Worthy reviewer, during formation of amorphous salts sometimes peaks intensities decreases while some peak disappears

  1. What culture medium was used for the antimicrobial experiments?

  • Bacteria cultures were grown in nutrient broth whereas in antibacterial tests nutrient agar has been used ( coli; MTCC 1687, S. typhi; MTCC 734, K. pneumonia; MTCC 1030).

  1. Why you only choose bacteria from Enterobacteriaceae family.
  • Worthy reviewer, the selection was based on availability of the strains. These were available and therefore have been used.

  1. What could be the effect of using a gram-positive bacterium in this study
  • Worthy reviewer, it is a broad-spectrum and Definity would have bactericidal effect. The selected antibiotic is a well-established broad-spectrum drug the strategy used here is to increase its bioavailability rather than expecting a different efficacy against the gram positive or negative strain.

  1. I would like the figures of the antibiogram test  
  • Worthy reviewer, here the strategy was to enhance bioavailablity through cocrystal formation rather than checking susceptibly of the drug towards the used bacterial strains. Levofloxacin is an established drug with broad-spectrum efficacy. Also, we do not have the facilities to perform the test suggested by worthy reviewer. In future we will endorse the valuable suggestion.

Reviewer 2 Report

The manuscript by Islam et al described the development of cocrystal and amorphous salt of levofloxacin hemihydrate (LEV) with phthalimide (PTH) and caffeic acid (CFA). The developed materials were characterized with the help of FT-IR, Raman spectroscopy, powder X-ray diffraction (PXRD), differential scanning calorimetry (DSC) and thermogravimetric analysis (TGA). The authors compared the dissolution rate and antimicrobial activity against selected strains, K. pneumonia, E. coli and S. typhi of parent drug and the new materials. It is proposed that these new formulations enhance the dissolution rate as well as antibacterial efficiency and could be potent against MDR bacterial strains.

The paper is interesting, but few issues need clarification and corrections:

It’s not entirely clear from the paper what are the factors that are responsible in improving the properties of  Levofloxacin hemihydrate in newly developed materials.

How author established the homogeneity in stoichiometric assembly for LEV-PTH and LEV-CFC?

The author should consider labeling the functional group frequencies in figure 2 for clarity. The quality of figures needs improvement.

Line 300: “..there is not much difference in the dissolution profile of the cocrystal and amorphous salt in simulated gastric fluid as compared with pure drug except 20 minutes time point …” any plausible explanation for this behavior?

Minor comments:

1. Line 39, “..APIs..”authors should first describe the full form before using abbreviations.

2.  Table 1, corresponding to phthalimide: what is 23? Looks like typo.

3.   Line 277, it should be 1H with 1 in superscript, same correction in the following paragraph.

Author Response

Reviewer 2

The manuscript by Islam et al described the development of cocrystal and amorphous salt of levofloxacin hemihydrate (LEV) with phthalimide (PTH) and caffeic acid (CFA). The developed materials were characterized with the help of FT-IR, Raman spectroscopy, powder X-ray diffraction (PXRD), differential scanning calorimetry (DSC) and thermogravimetric analysis (TGA). The authors compared the dissolution rate and antimicrobial activity against selected strains, K. pneumonia, E. coli and S. typhi of parent drug and the new materials. It is proposed that these new formulations enhance the dissolution rate as well as antibacterial efficiency and could be potent against MDR bacterial strains.

The paper is interesting, but few issues need clarification and corrections:

It’s not entirely clear from the paper what are the factors that are responsible in improving the properties of Levofloxacin hemihydrate in newly developed materials.

  • Worthy reviewer, for low solubility drugs the bioavailablity can be improved by converting them to nano size drug or making their cocrystals. Same thing has been attempted here. Taking into consideration your valuable suggestion some related literatures about cocrystal and salt of LEV are added in revised manuscript

How author established the homogeneity in stoichiometric assembly for LEV-PTH and LEV-CFC?

  • The molar ratio of LEV-PTH (1:1) and LEV-CFA (1:1) has been confirmed by NMR analysis.

The author should consider labeling the functional group frequencies in figure 2 for clarity. The quality of figures needs improvement.

  • Worthy reviewer of quality figure improved and labeled the functional group frequencies.

Line 300: “..there is not much difference in the dissolution profile of the cocrystal and amorphous salt in simulated gastric fluid as compared with pure drug except 20 minutes time point …” any plausible explanation for this behavior?

  • Worthy reviewer, may this have happen due to human error, the rest there is no specific reason.

Minor comments:

  1. Line 39, “..APIs..”authors should first describe the full form before using abbreviations.
  • Worthy reviewer, the said correction is made in revised manuscript.
  1. Table 1, corresponding to phthalimide: what is 23? Looks like typo.
  • Worthy reviewer, the typo mistake is corrected accordingly
  1. Line 277, it should be 1H with 1 in superscript, same correction in the following paragraph.
  • Worthy reviewer, Corrected accordingly

Round 2

Reviewer 2 Report

The author's have addressed the issue. I'm happy with this revision.